# Identification and Evolutionary Analysis of the Auxin Response Factor (ARF) Family Based on Transcriptome Data from Caucasian Clover and Analysis of Expression Responses to Hormones

**DOI:** 10.3390/ijms242015357

**Published:** 2023-10-19

**Authors:** Jingwen Jiang, Zicheng Wang, Zirui Chen, Yuchen Wu, Meiqi Mu, Wanting Nie, Siwen Zhao, Guowen Cui, Xiujie Yin

**Affiliations:** College of Animal Science and Technology, Northeast Agricultural University, Harbin 150030, China; jjw990824@163.com (J.J.); fzqwzc@163.com (Z.W.); czr990413@126.com (Z.C.); wuyuchen202211@163.com (Y.W.); mmq0616@163.com (M.M.); ronglan105284b@163.com (W.N.); zhaosiwen2023@163.com (S.Z.)

**Keywords:** Caucasian clover, auxin response factors, auxin, evolutionary analysis, hormonal response

## Abstract

Caucasian clover (*Trifolium ambiguum* M. Bieb.) is an excellent perennial plant in the legume family Fabaceae, with a well-developed rhizome and strong clonal growth. Auxin is one of the most important phytohormones in plants and plays an important role in plant growth and development. Auxin response factor (ARF) can regulate the expression of auxin-responsive genes, thus participating in multiple pathways of auxin transduction signaling in a synergistic manner. No genomic database has been established for Caucasian clover. In this study, 71 *TaARF* genes were identified through a transcriptomic database of Caucasian clover rhizome development. Phylogenetic analysis grouped the TaARFs into six (1–6) clades. Thirty TaARFs contained a complete ARF structure, including three relatively conserved regions. Physical and chemical property analysis revealed that TaARFs are unstable and hydrophilic proteins. We also analyzed the expression pattern of *TaARFs* in different tissues (taproot, horizontal rhizome, swelling of taproot, rhizome bud and rhizome bud tip). Quantitative real–time RT–PCR revealed that all *TaARFs* were responsive to phytohormones (indole-3-acetic acid, gibberellic acid, abscisic acid and methyl jasmonate) in roots, stems and leaves. These results helped elucidate the role of ARFs in responses to different hormone treatments in Caucasian clover.

## 1. Introduction

Caucasian clover (*Trifolium ambiguum* M. Bieb.) is a perennial legume forage. It is native to the cold Caucasus mountains, eastern Turkey and northern Iran. It is the only good forage in the legume clover genus with strong clonal growth from rhizomes and has high yield and forage value in both single and mixed sowings [1,2]. It has a well-developed root system, and the main root is clearly expanded and penetrates deep into the soil; it can grow into new plants through the rhizome and has a high regeneration capacity [3,4]. Studies have also shown that Caucasian clover is more resistant to cold and heat, has better drought resistance and is also more resistant to disease than other similar plants [5,6,7,8,9,10]. However, the low seed yield and weak seedling vigor of Caucasian clover make direct sowing for grassland restoration very difficult. This is also a key factor limiting the widespread use of Caucasian clover [11]. This limitation can be overcome by using its well-developed underground root tillers, which can be established by transplanting from rhizomes [12]. Many studies have also been conducted on the characteristics of rhizome development in Caucasian clover and the effect of phytohormone treatment and fertilization on rhizome development in Caucasian clover [13,14,15,16]. Therefore, it is important to study the regulatory mechanisms of phytohormones in Caucasian clover in relation to rhizome and leaf development.

Auxin is one of the most important phytohormones in plants and plays a key role in different developmental stages of plants, such as organogenesis, vascular tissue differentiation and apical dominance, and, at the cellular level, in most of the major growth responses in cells during processes including elongation, division and differentiation [17,18,19,20]. Auxin regulates a variety of genes, among which the more studied ones are those related to *auxin/indole-3-acetic acid (Aux/IAAs)*, *Gretchen Hagen 3 (GH3s)* and *Small auxin up RNAs (SAURs)* [21,22,23,24,25]. These primary/early auxin-responsive genes must contain the TGTCTC growth response element (AuxRE) in their promoters [26]. Auxin response factor (ARF) regulates the expression of auxin-responsive genes by specifically binding to auxin response elements in the promoter regions of these auxin-responsive genes, thus participating synergistically in multiple pathways of auxin transduction signaling [27].

When the external auxin concentration is low, the Aux/IAA (auxin/indole acetic acid) protein forms a dimer with the ARF protein to inhibit the ARF protein’s activity, preventing it from regulating the expression of downstream genes and disrupting the auxin signaling pathway. When the external auxin concentration increases, the auxin signal is sensed by the auxin receptor protein TIR1/AFB, which in turn transmits the signal to the cell to degrade the Aux/IAA protein and release the ARF protein, which binds to the promoter of the downstream gene to regulate the expression of the downstream gene [26,28].

The ARF protein consists of three relatively conserved regions: the first is an N-terminal DNA-binding domain (DBD) that binds to a specific conserved sequence (TGTCTC) in the AuxREs, but the DBD region itself does not respond to auxin [29]. The second is the middle region (MR) with an activation domain (AD) or repression domain (RD). The activation or repression effect of ARFs during transcription depends on the amino acid composition of the MR region, with the MR region rich in glutamine (Q-rich) being the transcriptional activation region (AD) and the region rich in proline, serine or threonine (P/S/T-rich) being the transcriptional repression region (RD) [30]. The third domain is the carboxy-terminal dimerization domain (CTD), which is structurally homologous to the repressor protein Aux/IAA and acts as a transcriptional regulator [31,32,33,34].

ARF proteins have been widely explored in diverse species because of their important roles in various physiological and biological processes. *ARF1* was cloned for the first time in *Arabidopsis thaliana*, thanks to the identification of the AuxRE sequence [35]. An increasing number of members of the ARF gene family have since been mined in various species, such as rice [36], corn [37,38], tomato [39], apple [40] and *Populus trichocarpa* [41]. Many ARF gene family members have also been identified in legume family members such as soybean, bean [42] and *Medicago truncatula* [43]. Thus, the function of the *ARF* gene is gradually being discovered. Many studies have shown that *ARF* genes play a role in plant root development. Shorter lateral root length and increased density were observed in *Medicago truncatula* when *MtARF2*, *MtARF3*, *MtARF4a* and *MtARF4b* expression levels were reduced [44]. In *Arabidopsis*, *AtARF7* and *AtARF19* are involved in inducing hypocotyl adventitious root development. The *Arabidopsis* mutants *arf7* and *arfF19* show lateral root deficiency, which can be restored when maize *ZmARF4* is overexpressed in the mutant. *ZmARF4* overexpression in *Arabidopsis* significantly increased root length, root volume and root tip number [45].

In the present study, based on full-length transcriptome data of the roots of Caucasian clover, we identified all 71 members of the TaARF gene family and performed systematic analysis of the basic characteristics of the poplar ARF family. The expression patterns of TaARF genes were analyzed with transcriptomic data generated from different organs and at different developmental stages of Caucasian clover. In addition, we performed quantitative real-time PCR (qRT–PCR) analysis on the roots, stems and leaves of Caucasian clover under different hormone treatments. Our results lay an important foundation for future research.

## 2. Results

### 2.1. Transcriptome-Wide Identification of ARFs in Caucasian Clover

In this study, we first used the B3 (PF02362) and Auxin_resp (PF06507) domains to perform a hidden Markov model search on our transcriptome data from Caucasian clover. A total of 89 candidate ARF proteins were obtained. Next, we verified the reliability of the screened candidate ARF members. We used SMART and NCBI-CDD to determine the integrity of these domains of the candidate protein. Finally, we removed redundant and irrelevant genes and obtained 71 ARF gene family members. For convenience, we named the 71 *ARFs ARF1-ARF71* according to their transcriptome IDs, by referring to Yang et al. [46]. In addition, we analyzed the physicochemical properties of these 71 TaARF proteins (Appendix A). The proteins encoded by the 71 TaARFs contained 306 (TaARF22) to 1158 (TaARF65) amino acids, with molecular weights (MWs) ranging from 34,129.98 Da (TaARF22) to 129,038.8 Da (TaARF34). The isoelectric points (pIs) of these proteins ranged from 5.4 (TaARF10) to 9.58 (TaARF2). The instability index analysis showed that all ARF proteins were unstable proteins. The aliphatic index (AI) was between 58.46 (TaARF44) and 84.43 (TaARF40). The grand average of hydropathicity (GRAVY) results showed that the hydrophobicity indices of the ARF proteins were all less than 0, and thus, they were hydrophilic proteins.

### 2.2. Phylogenetic Analysis of TaARFs

To investigate the evolutionary relationships of TaARFs, we used 22 AtARF proteins from *Arabidopsis* [45], 22 MtARF proteins from *Medicago truncatula* [47], and 71 TaARF proteins from Caucasian clover to construct a phylogenetic tree using the maximum likelihood method. The phylogenetic tree of 114 genes was divided into six clades (clade 1, clade 2, clade 3, clade 4, clade 5 and clade 6) (Figure 1). Clade 2 did not contain TaARFs and MtARFs. Genes in the same clade may have similar functions. The number of ARF genes in each clade suggests significant interspecific differences in the ARF gene family between Caucasian clover and *Arabidopsis*, but some similarities in the ARF gene family between Caucasian clover and *Medicago truncatula*. 

### 2.3. Transcriptome-Wide Identification of ARFs in Caucasian Clover

The gene domains and the composition of conserved motifs of 71 TaARF proteins were analyzed and are shown according to their phylogenetic relationships (Figure 2A–C). The phylogenetic tree was constructed with the maximum likelihood method, using 1000 bootstrap replicates, for the 71 full-length proteins in MEGA7.0 (Figure 2A). A conserved motif search performed on 71 TaARF proteins through the Maximization for Motif Elicitation (MEME) online website detected 10 conserved motifs (motifs 1–10; Figure 2B). Meanwhile, we analyzed the gene domains of TaARFs and visualized the coordinates of the conserved domains of ARFs directly mapped to the genetic structure of Caucasian clover (Figure 2C). The results showed that motif 1, motif 3, motif 4, motif 6, motif 9 and motif 10 are present in all TaARF proteins, except TaARF40, which does not contain motif 4. All thirty genes containing the CTD structural domain contained motif 5. Only five genes did not contain motif 2. The 71 TaARF proteins have a high degree of similarity and are relatively evolutionarily conserved. The sequence identity of the conserved pattern of TaARF structural domain proteins identified with Multiple Expectation Maximization for Motif Elicitation (MEME) program (http://memesuite.org/tools/meme, accessed on 15 September 2022) is shown in Appendix A. Gene domain analysis showed that all TaARFs contained the B3 domain and the Auxin_resp domain, but only 30 TaARFs contained the CTD domain.

### 2.4. Expression Patterns of TaARF Genes in Different Tissues of the Caucasian Clover Rhizome

To explore the potential functions of TaARFs in Caucasian clover rhizome development, we used RNA-seq data to analyze the transcript levels of TaARFs in the taproot (ZG), horizontal rhizome (SP), swelling of the taproot (PD), rhizome bud (YY) and rhizome bud tip (YJ) (Figure 3).

The results showed that a total of eleven *TaARF (TaARF5/8/9/10/15/25/26/52/53/59/63)* genes were highly expressed in the swelling of the taproot; 10 TaARF *(TaARF2/4/20/21/29/34/44/54/55/60)* genes were highly expressed in horizontal rhizomes; twenty-one *TaARF* genes *(TaARF2/7/26/32/33/36/37/38/43/45/48/49/56/57/63/67/68/69/70)* were more highly expressed in the rhizome bud tip; six genes *(TaARF23/29/40/64/69/71)* were more highly expressed in the taproot; and only twenty-four genes were less expressed in the rhizome bud. *TaARF2* was highly expressed in three tissues: horizontal rhizome, rhizome bud and rhizome bud tip; *TaARF7/14/32/33/36/37/38/49/57* was highly expressed in two tissues, namely the rhizome bud and rhizome bud tip. *TaARF5/8/9/10/15/25/26/52/53/59/63* had higher expression in the swelling of the taproot; *TaARF23/29/64/69/71* had higher expression in the taproot. *TaARFs* are functionally diverse, with more genes functioning in Caucasian clover rhizome buds and rhizome bud tips, and a few genes functioning in Caucasian clover taproots and horizontal rhizomes. 

### 2.5. Expression Analysis of TaARF Genes in Response to Hormones

Treatment of Caucasian clover with IAA showed that all *TaARFs* responded to IAA, with *TaARF11/23* showing a decreasing then increasing trend in roots; *TaARF37* showing an increasing, then decreasing and then increasing trend in roots; *TaARFs* showing an increasing, then decreasing and then increasing trend in leaves; and four *TaARFs (TaARF6/10/11/18)* showing significantly decreasing expression in leaves (*p* < 0.05) (Figure 4). *TaARF15* was expressed in higher amounts in roots than in leaves, in contrast to the other genes. *TaARF6/10/18/23* expression was highest in leaves after 3 h of treatment and in roots after 12 h of treatment.

The expression of *TaARF6/10/11/18* was also significantly higher in leaves (*p* < 0.05) and higher than in roots and stems when treated with ABA in Caucasian clover. The expression of *TaARF6/10/15/18/21/23* in stems showed an increasing trend with an increasing duration of ABA treatment, while the expression of *TaARF10/18* in stems showed a decreasing trend first. Only two *TaARFs (21/23)* showed decreased and then increased expression in the roots, while all other *TaARFs* showed increased expression at 3 h (Figure 5).

When we treated Caucasian clover with GA3, *TaARF15/23/37* expression was higher in roots than in stems and leaves at one time, *TaARF6/10/16* expression was higher in stems than in roots and leaves and *TaARF6/10* expression levels in stems showed the same trend (Figure 6).

In MeJA-treated Caucasian triticale, *TaARF6/15/11* expression in leaves was the same as that in the other hormone treatments and remained high. Notably, *TaARF21/23* expression was increased in the stems after MeJA treatment, which is different from the results observed in the other hormone treatments (Figure 7).

## 3. Discussion

Auxins play a very important role in the regulation of plant growth and development [48]. The signal transduction of auxins controls the responses of organs and tissues to auxins and plays an important role in the regulation of plant growth by auxins [18,49,50,51,52]. As transcription factors, ARFs participate in the signaling pathways related to the auxin response and regulate the growth and development of plants [53]. ARF genes have been continually reported in many different plant species, but they have not been reported within Caucasian clover. In our studies, we identified a total of 71 TaARF genes, with more TaARF family members than *Arabidopsis* (23), tomato (17), rice (25), apple (31), poplar (35) and *Tribulus terrestris* (22) [36,40,42,43,54]. ARF members were more abundant in soybean, peanut and blueberry, with 51, 61 and 60, respectively [52,55,56]. This suggests that the ARF gene family has undergone considerable duplication and diversification in Caucasian clover.

Physicochemical analysis of these 71 TaARFs revealed that 40 TaARFs with a pI less than seven functioned in an acidic subcellular environment, TaARF31/58 with a pI equal to seven functioned in a neutral subcellular environment, and the few TaARFs with a pI greater than seven functioned in a basic subcellular environment. All TaARFs are unstable proteins, and a negative GRAVY value indicates that all TaARFs are hydrophilic proteins (Appendix A).

Phylogenetic analysis revealed that AtARFs were divided into six clades, with TaARFs and MtARFs distributed in five of them. The second clade contained only AtARFs. We noted that no MtARF proteins clustered with AtARF12–15 or AtARF21–23. Some studies have shown that these genes form a cluster on *Arabidopsis* chromosome I and are mainly expressed during embryogenesis and seed development [57]. Therefore, there are no genes similar to *AtARF12–15 or AtARF21–23* in Caucasian clover and *Medicago truncatula*. Additionally, in *Magnolia sieboldii* and *Oryza sativa*, the same result was reported [58]. In addition, the phylogenetic tree shows that the phylogenetic relationship between Caucasian clover and *Medicago truncatula* is more conserved than that between Caucasian clover and *Arabidopsis*, probably because Caucasian clover and *Medicago truncatula* belong to the same family, and indicates that the ARF genes are relatively conserved but also different between species, and they may have relatively conserved molecular functions [59].

Gene domain analysis reveals the functional characteristics of proteins. The results of gene domain analysis showed that all 71 TaARFs contained the B3 domain and the ARF-specific Auxin_resp domain, which are essential for the ARF gene family [26,35]. The CTD domain can form a heterodimer with Aux/IAA or a homodimer with ARF, and a total of 41 TaARFs in this study did not contain this domain [60]. ARFs found to lack the CTD structural domain in tomato, rice and wheat may be regulated in a growth factor-independent manner [36,39,61]. Using conserved motifs and gene domain analyses, we found that motif 10 in TaARFs may correspond to CTD structural domains. TaARFs are more developmentally conserved, but motifs and domains of the same branch appear to differ, reflecting the diversity of genes.

Expression pattern analysis can help us to screen *TaARF* candidate genes with different potential functions [62]. Analysis of the expression pattern of Caucasian triticale rhizomes showed that more *TaARFs* were expressed in YY and YJ than in the other three tissues, suggesting that these genes may function in the rhizome bud, with only a small number of *TaARFs* functioning in the taproot, swelling of the taproot and horizontal roots. It has been shown that *AtARFs* are involved in the regulation of plant morphological growth; for example, ARF7 and ARF19 double knockout mutants show severe impairment of lateral root formation in *Arabidopsis* [63].

*ARFs* not only regulate growth hormone gene expression but are also involved in other hormonal regulatory pathways, such as the gibberellin, abscisic acid (ABA), ethylene, oleuropein lactone and salicylic acid pathways [64,65,66]. Many *AtARFs* are also involved in hormone signaling; for example, *AtARF2* and *AtARF19* are considered key genes in the auxin and ethylene signal transduction pathways, and *AtARF6* and *AtARF8* regulate the expression of *JAZ/TAFY 10A*, which is controlled by jasmonic acid (JA) [36,63,67]. We therefore selected one *TaARF* member from each clade of the phylogenetic tree and treated Caucasian clover with different hormones to investigate the function of *TaARFs*. We found that these *TaARFs* responded to all of these hormones. When we treated these *TaARFs* with IAA, ABA, GA3, and MeJA, the trend in the rise and fall of *TaARF* expression was not uniform. This pattern also occurs in hairy poplar, *Nitraria sibirica* Pall, eggplant, peanut and other plants [54,56,68,69].

## 4. Materials and Methods

### 4.1. Identification and Sequence Analysis of ARF Genes from Caucasian Clover

To identify ARF gene family members, we searched the full-length transcriptome database of Caucasian clover rhizomes (accession number: PRJNA586585) and downloaded ARF conserved structural domain B3 (PF02362) and Auxin_resp (PF06507) from the Pfam database. We performed an HMM search of the Caucasian triticale protein data to initially screen for ARF gene family members. Redundant sequences were then manually removed, and all candidate sequences were further checked using the Pfam (http://pfam.sanger.ac.uk/search, accessed on 1 September 2022) and SMART (http://smart.embl-heidelberg.de/, accessed on 1 September 2022) databases.

### 4.2. Physicochemical Properties and Conserved Motif Analysis

The ExPASy website was used to calculate the pI, MW, GRAVY and amino acid length of the ARF protein, and subcellular localization was forecasted by the online software Cell-PLoc-2.0. The online Multiple Expectation Maximization for Motif Elicitation (MEME) program (http://memesuite.org/tools/meme, accessed on 15 September 2022) was used to analyze the conserved motif structures of the proteins encoded by the TaARFs (parameter settings: maximum number of motifs, 10; maximum width, 50).

### 4.3. Multiple Sequence Alignment and Phylogenetic Analysis

The identified Caucasian clover ARF protein sequences were aligned with the ARF gene family protein sequences of *Arabidopsis* and alfalfa using MUSCLE, and evolutionary trees were constructed using the maximum likelihood method in MEGA 7.0 software (bootstrapping: 1000 replications). The evolutionary trees were beautified using the online website EvolView (https://evolgenius.info//evolview-v2/#login/) accessed on 13 October 2022.

### 4.4. Expression Analysis of Different Tissue Sites

The transcriptomic data generated from different organs and for different developmental stages of Caucasian clover have been described previously [70]. Based on the sequencing data, the differential expression of the ARF gene family at different sites was analyzed. The expression data of ARF family genes in the taproot, horizontal rhizome, swelling of the taproot, rhizome bud and rhizome bud tip were analyzed by the logarithmic (log) calculation method. The expression volume was visualized by TBtools [71].

### 4.5. Plant Materials

The experimental materials were collected from Caucasian clover plants in good condition planted in the greenhouse at Northeast Agricultural University. The greenhouse had a day/night average temperature of 24 °C, a photoperiod of 16/8 h (light/dark) and a relative humidity of 70–80%. The vermiculite was kept moist with 1/2-strength Hoagland nutrient solution. Caucasus clover seedlings cultured for 35 d were subjected to the following hormone treatments: IAA 1 mmol/L, ABA 1 mmol/L, MeJA 1 mmol/L and GA3 2 mmol/L. Samples were taken at 0, 3, 6, 12, 24 and 48 h after each treatment. The treatment with no hormone was used as a control (Control), and each treatment had three replicates. Samples were taken from the roots, stems and leaves of each treatment, wrapped in aluminum foil, quickly frozen in liquid nitrogen, and then stored at −80 °C.

### 4.6. RNA Extraction, cDNA Synthesis and Quantitative RT–PCR

Total RNA was extracted using an Ultrapure RNA Kit (CWBIO, Taizhou, China). The cDNA template for quantitative RT–PCR was synthesized using HiScript II Reverse Transcriptase (Vazyme, Nanjing, China). With cDNA used as a template, real-time fluorescence quantification was performed using internal reference and fluorescence quantification primers (Appendix A). All qRT–PCR analyses were performed using a ChanQ Universal SYBR qPCR Master Mix Kit according to the manufacturer’s instructions, and the relative gene expression was calculated using the 2^−∆∆Ct^ method. The qRT–PCR methods are listed in Appendix A. All reactions were performed with three replicates. SPSS software was used for significance analysis (*p* < 0.05) [72].

### 4.7. Statistical Analysis

Excel 2019 was used to organize the data, and mapping was performed with Origin software 2019b and TBtools. One-way analysis of variance was performed via SPSS 23.0, and Duncan’s method was used for multiple comparisons.

## Figures and Tables

**Figure 1 ijms-24-15357-f001:**
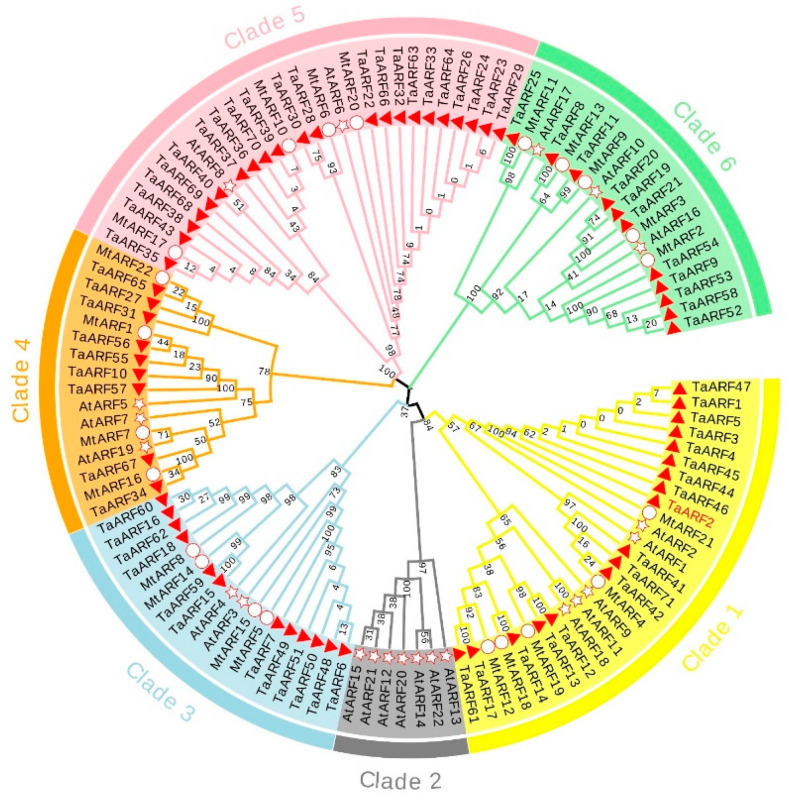
Phylogenetic tree of ARF proteins from Caucasian clover and *Arabidopsis*. A phylogenetic tree of the ARF gene family was constructed by MEGA 7 software using the maximum likelihood (ML) option with 1000 bootstrap replicates. Red triangles, red circles and red stars indicate Caucasian clover, *Medicago truncatula* and *Arabidopsis*, respectively. Differently colored branches represent different ARF subfamilies.

**Figure 2 ijms-24-15357-f002:**
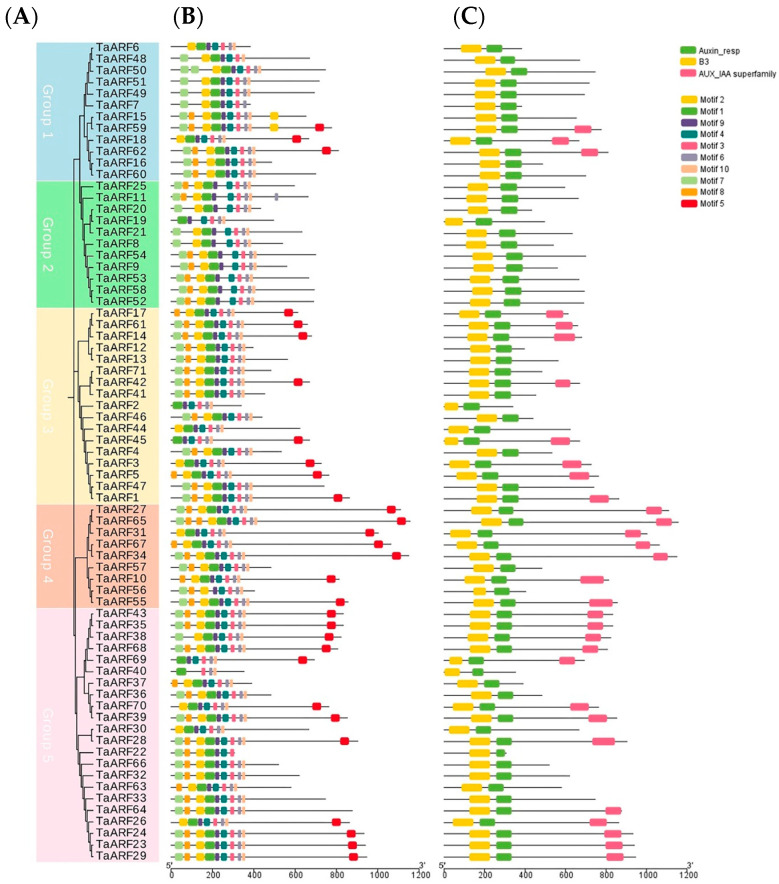
The phylogenetic relationship, conserved motifs and gene domains of TaARFs. (**A**): The maximum likelihood (ML) phylogenetic tree of TaARF proteins, constructed using full-length sequences with 1000 bootstrap replicates; (**B**): distribution of conserved motifs in TaARF proteins. A total of 10 motifs were predicted, and the scale bar represents 100 aa. (**C**): Distribution domains of TaARFs.

**Figure 3 ijms-24-15357-f003:**
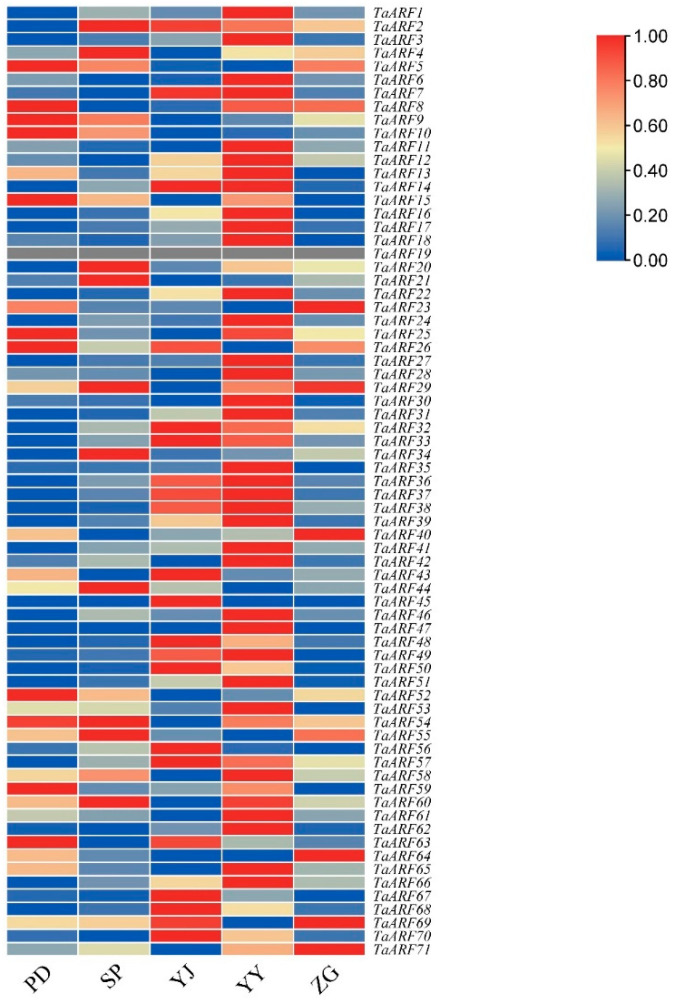
Heatmap showing the *TaARF* gene expression pattern of Caucasian clover. Note: A, expression pattern of *TaARF* genes in different tissues; ZG, taproot; SP, horizontal rhizome; PD, swelling of taproot; YY, rhizome bud; YJ, rhizome bud tip.

**Figure 4 ijms-24-15357-f004:**
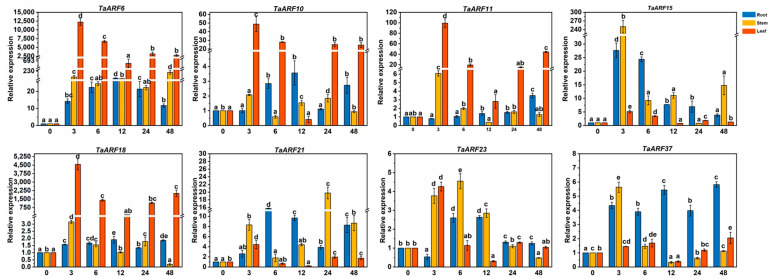
Expression analysis of *TaARFs* after IAA treatment. The relative mRNA levels of the group at 0 h were used as a reference. The relative expression was calculated using the 2^−∆∆Ct^ method. Three biological replicates were carried out for each experiment. The different lowercase letters indicate significant differences under different treatment times for the same tissue.

**Figure 5 ijms-24-15357-f005:**
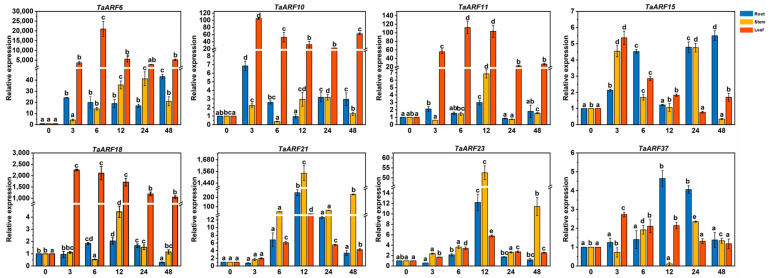
Expression analysis of *TaARFs* after ABA treatment. The relative mRNA levels of the group at 0 h were used as a reference. The relative expression was calculated using the 2^−∆∆Ct^ method. Three biological replicates were carried out for each experiment. The different lowercase letters indicate significant differences under different treatment times for the same tissue.

**Figure 6 ijms-24-15357-f006:**
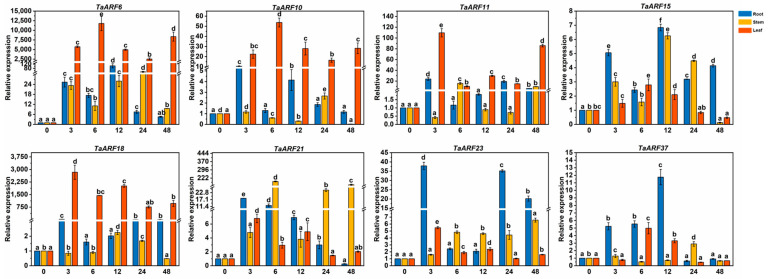
Expression analysis of *TaARFs* after GA3 treatment. The relative mRNA levels of the group at 0 h were used as a reference. The relative expression was calculated using the 2^−∆∆Ct^ method. Three biological replicates were carried out for each experiment. The different lowercase letters indicate significant differences under different treatment times for the same tissue.

**Figure 7 ijms-24-15357-f007:**
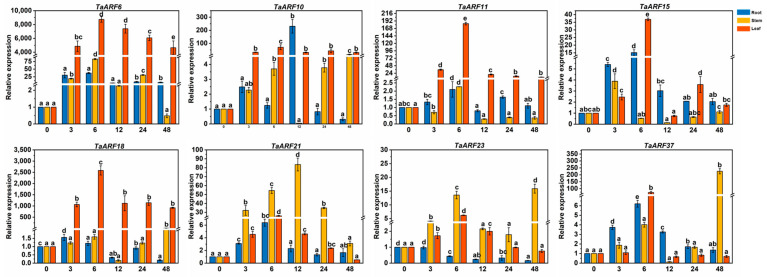
Expression analysis of *TaARFs* after MeJA treatment. The relative mRNA levels of the group at 0 h were used as a reference. The relative expression was calculated using the 2^−∆∆Ct^ method. Three biological replicates were carried out for each experiment. The different lowercase letters indicate significant differences under different treatment times for the same tissue.

## Data Availability

Raw reads of full-length transcriptome data from this study are available as a BioProject from the National Center for Biotechnology Information (NCBI) (https://www.ncbi.nlm.nih.gov/bioproject/, accessed on 4 September 2023) under access number PRJNA586585.

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
