# Peer review of "Identification and Evolutionary Analysis of the Auxin Response Factor (ARF) Family Based on Transcriptome Data from Caucasian Clover and Analysis of Expression Responses to Hormones"

_ijms, 2023, doi:10.3390/ijms242015357_

Round 1
Reviewer 1 Report
The authors tried to identify and study the response of auxin response factor (ARF) family genes from Trifolium ambiguum to treatment with various auxins. They identified TaARF genes using available a transcriptome data. Although the manuscript consists of interesting data, the presentation requires a neat revision.
1. Language is not very clear.
2. The title has some grammar mistake. As well as it is very confusing. Revise the title.
3. Avoid using abbreviations in the abstract.
4. Similarly expand all abbreviations at their first appearance (eg. IAA/ GH3 in line 47). Check throughout the manuscript.
4. italicise the scientific names (Eg. “Arabidopsis thaliana” in line number 120)
5. while representing the qRT-PCR expression analysis formula represent −ΔΔCt as superscript (2−ΔΔCt).
6. which statistical method was used to find the significance in qRT-PCR expression analysis? (provide the method in line number 296)
7. version of SPSS software used is missing.
8. in line number 289, “for reverse transcription PCR”? RNA is the template for cDNA synthesis. Revise it as “for qRT-PCR”
9. “Mapping with Origin software” in line number 296. Whether the bar graph was created using this software?
10. rearrange the materials and methods section in correlation with results section.
The clarity of the language could be improved. A thorough grammar check is required after all corrections related to the flow of the article is made.
Author Response
For research article
Identification and evolutionary analysis of the Auxin Response Factor (ARF) family based on transcriptome data from Cauca-sian clover and analysis of expression responses to hormones
Response to Reviewer 1 Comments
|
||
Summary |
|
|
Thank you very much for the careful review and constructive suggestions regarding our manuscript, entitled “Identification and evolutionary analysis of the Auxin Response Factor (ARF) family based on transcriptome data from Cauca-sian clover and analysis of expression responses to hormones”. All the comments were valuable and very helpful for revising and improving our paper and for providing important guidance for our research. We have studied these comments carefully and tried our best to revise and improve the manuscript accordingly. We used blue for changed text in the revised manuscript. Thank you again for all your help, and we look forward to hearing from you soon.
|
||
Point-by-point response to Comments and Suggestions for Authors Comment 1: Avoid using abbreviations in the abstract. Responses 1: Thank you for your suggestion. We have replaced all abbreviations in the abstract.
Comment 2: Similarly expand all abbreviations at their first appearance (eg. IAA/ GH3 in line 47). Check throughout the manuscript. Responses 2: We provided the full names of all the abbreviations at first mention, such as auxin/indole-3-acetic acid (Aux/IAAs), Gretchen Hagen 3 (GH3s), small auxin up RNAs (SAURs), molecular weights (MWs) and isoelectric points (pIs).
Comment 3: italicise the scientific names (Eg. “Arabidopsis thaliana” in line number 120). Responses 3: Based on your suggestion, we have italicized all species names in the article.
Comment 4: while representing the qRT-PCR expression analysis formula represent −ΔΔCt as superscript (2−ΔΔCt). Response 4: We have changed −ΔΔCt to a superscript.
Comment 5: which statistical method was used to find the significance in qRT-PCR expression analysis? (provide the method in line number 296). Response 5: For significance in qRT‒PCR expression analysis, Duncan's method was used. We have provided information on the software used for data sorting, statistical analysis and graphing in section 4.7. 4.7 Statistical analysis Excel 2019 was used to organize the data, and mapping was performed with Origin software and TBtools. One-way analysis of variance was performed via SPSS 23.0, and Duncan’s method was used for multiple comparisons.
Comment 6: version of SPSS software used is missing.. Responses 6: The SPSS software version (23.0) has been added.
Comment 7: in line number 289, “for reverse transcription PCR”? RNA is the template for cDNA synthesis. Revise it as “for qRT-PCR” Responses 7: Thank you for your professional comments on our article. We have changed the text to "The cDNA template for quantitative RT‒PCR was synthesized using HiScript II Reverse Transcriptase."
Comment 8: Mapping with Origin software” in line number 296. Whether the bar graph was created using this software Responses 8: Yes, the bar chart in this article was created using Origin software.
Comment 9: rearrange the materials and methods section in correlation with results section Responses 9: Based on your valuable suggestions, we have revised the order of the materials and methods sections. The current order is as follows: 4.1. Identification and sequence analysis of ARF genes from Caucasian clover, 4.2 Physicochemical properties and conserved motif analysis, 4.3 Multiple sequence alignment and phylogenetic analysis, 4.4 Expression analysis of different tissue sites, 4.5 Plant materials, 4.6 RNA extraction, cDNA synthesis and quantitative RT‒PCR, and 4.7 Statistical analysis. The order is partially consistent with that of the results.
|
||
Response to Comments on the Quality of English Language |
||
Point 1: Language is not very clear.
|
||
Response 1: Thank you for your comment. We have invited several native English speakers to help polish the language in our article. We hope that the revised manuscript is acceptable. |

Reviewer 2 Report
Manuscript number: ijms-2622320299
Manuscript title: Identification and responding to exogenous hormone of Auxin Response Factor (ARF) family based on transcriptome data of Caucasian clover
Authors: Jingwen Jiang, Zichen Wang, Zirui Chen, Yuchen Wu, Meiqi Mu, Wanting Nie, Siwen Zhao, Xiujie Yin, and Guowen Cui 1
In the present manuscript, transcription factors interacting with the promoters of auxin-responsive genes were studied in Caucasian clover at the phylogenetic, molecular and biochemical levels. The treated topic is of interest for the Journal readership, the amount of the experimental work done is considerable, the methodology appears to be adequate. The statistical treatment of the experimental data seems to be adequate as well.
1) The very main problem I see here, before the present manuscript can be further considered for publication, has to do with clarity, care in editing, and the use of English. Indeed, it needs to be re-written from its very beginning, paying much attention to:
- grammar
- editing
- typing errors
- truncated sentences
- redundancy
- pleonasms (e.g. lines 58-59)
- repetition (for the first time in my life as a reviewer I found the content of two sub-headings to be identical to each other, see 2.2 and 2.3!)
- sense and consequence
If I were the Authors, I would take such recommendations quite seriously, because it is not a matter of mere punctual correction and/or re-styling, here. The present manuscript needs to be re-written by a competent writer in English, or even better thoroughly checked by a professional editing service.
2) The fonts used for axes and legends in Figures 4-6 are illegible, because too small
3) I would like the Authors to expand/comment more in depth (within their revised version) their statement appearing at the very end of the Introduction (line 92), i.e. “Our results lay an important foundation for future.” In which sense, please? It may be that this statement is true and that I am unable to fully appreciate the implications and consequences of the present study, which, indeed, appears to me to be too much descriptive at times, and too little dedicated to envisage the functional implications, if any, and/or the potential applications, if any, of the results obtained. ..Providing either of the two, or both, would be made convincing to the reader, then the statement on line 92 would become fully reasonable and justified.
All the above considering, I recommend major revision of the present manuscript, convincingly and carefully addressing all the points raised above.
The present manuscript needs to be re-written by a competent writer in English, or even better thoroughly checked by a professional editing service.
Author Response
For research article
Identification and evolutionary analysis of the Auxin Response Factor (ARF) family based on transcriptome data from Cauca-sian clover and analysis of expression responses to hormones
Response to Reviewer 2 Comments
|

Round 2
Reviewer 1 Report
The authors significantly revised the manuscript. They answered all my comments. I have one more comment,
* Figure 2 subsections (A,B,C) is not mentioned in the image. Only it appears in the figure legend.
Author Response
For research article
Identification and evolutionary analysis of the Auxin Response Factor (ARF) family based on transcriptome data from Cauca-sian clover and analysis of expression responses to hormones.
Response to Reviewer 1 Comments
|
||
1. Summary |
|
|
Thank you very much for the careful review and constructive suggestions regarding our manuscript, entitled “Identification and evolutionary analysis of the Auxin Response Factor (ARF) family based on transcriptome data from Cauca-sian clover and analysis of expression responses to hormones”. All the comments were valuable and very helpful for revising and improving our paper and for providing important guidance for our research. We have studied these comments carefully and tried our best to revise and improve the manuscript accordingly. We used blue for changed text in the revised manuscript. Thank you again for all your help, and we look forward to hearing from you soon.
|
||
2. Questions for General Evaluation |
Reviewer’s Evaluation |
Response and Revisions |
Does the introduction provide sufficient background and include all relevant references? |
Yes |
|
Are all the cited references relevant to the research? |
Yes |
|
Is the research design appropriate? |
Can be improved |
|
Are the methods adequately described? |
Can be improved |
|
Are the results clearly presented? |
Can be improved |
|
Are the conclusions supported by the results? |
Yes |
|
3. Point-by-point response to Comments and Suggestions for Authors |
||
Comments 1: Figure 2 subsections (A,B,C) is not mentioned in the image. Only it appears in the figure legend.
|
||
Response 1: Thank you for pointing this out. I agree with this comment. Therefore, I have highlighted the subsections (A,B,C) in Figure 2. |
||
4. Response to Comments on the Quality of English Language |
||
|
||
|
||
5. Additional clarifications |
||
|
Reviewer 2 Report
Manuscript number: ijms-2622320 v2
Manuscript title: Identification and evolutionary analysis of the Auxin Response Factor (ARF) family based on transcriptome data from Caucasian clover and analysis of expression responses to hormones
Authors: Jingwen Jiang, Zichen Wang, Zirui Chen, Yuchen Wu, Meiqi Mu, Wanting Nie, Siwen Zhao, Guowen Cui, and Xiujie Yin
In the present revision version, I found that, respect to the original submission, the problems related to the use of English have been resolved.
Not the same, unfortunately, as far as completeness and clarity are concerned;
1) I wonder why Figures 4, 6, and 7 are never cited, or referred to, in the text. This of course implies that it is not easy for the reader to understand the correspondence among what is said in the text, and the figures in which the results are reported. This, which is not a trivial omission, points once more to inaccuracy and hastiness…A recurring problem in the present manuscript…
2) on this same vein, by comparing the original submission to the present revised version, one discovers that the colour legends, indicating the different tissue sources (root, stem, leaf) in Figures 4-6, have disappeared…Neither they have been replaced by an explanation whatever, either in the figures legends or in the text…Q. E. D….
All the above considering, I recommend major revision of the present manuscript, addressing the points raised above.
Author Response
For research article
Identification and evolutionary analysis of the Auxin Response Factor (ARF) family based on transcriptome data from Cauca-sian clover and analysis of expression responses to hormones.
Response to Reviewer 2 Comments
|
||
1. Summary |
|
|
Thank you very much for the careful review and constructive suggestions regarding our manuscript, entitled “Identification and evolutionary analysis of the Auxin Response Factor (ARF) family based on transcriptome data from Cauca-sian clover and analysis of expression responses to hormones”. All the comments were valuable and very helpful for revising and improving our paper and for providing important guidance for our research. We have studied these comments carefully and tried our best to revise and improve the manuscript accordingly. We used blue for changed text in the revised manuscript. Thank you again for all your help, and we look forward to hearing from you soon.
|
||
2. Questions for General Evaluation |
Reviewer’s Evaluation |
Response and Revisions |
Does the introduction provide sufficient background and include all relevant references? |
Yes |
|
Are all the cited references relevant to the research? |
Yes |
|
Is the research design appropriate? |
Yes |
|
Are the methods adequately described? |
Yes |
|
Are the results clearly presented? |
Must be improved |
|
Are the conclusions supported by the results? |
Yes |
|
3. Point-by-point response to Comments and Suggestions for Authors |
||
Comments 1: I wonder why Figures 4, 6, and 7 are never cited, or referred to, in the text. This of course implies that it is not easy for the reader to understand the correspondence among what is said in the text, and the figures in which the results are reported. This, which is not a trivial omission, points once more to inaccuracy and hastiness…A recurring problem in the present manuscript… |
||
Response 1: Thank you for your comment. Paragraphs 1-4 of “2.5 Expression analysis of TaARF genes in response to hormones” in manuscript correspond to Figures 4-7 respectively and are now marked in the manuscript.
|
||
Comments 2: by comparing the original submission to the present revised version, one discovers that the colour legends, indicating the different tissue sources (root, stem, leaf) in Figures 4-6, have disappeared…Neither they have been replaced by an explanation whatever, either in the figures legends or in the text…Q. E. D…. |
||
Response 2: Thank you for pointing this out. I agree with this comment. In the first manuscript submitted,the fonts used for axes and legends in Figures 4-7 are illegible. I modified Figure 4-7. In the modified picture, the colour legends, indicating the different tissue sources (root, stem, leaf) were placed in the upper right corner of the Figures |
||
|
||
4. Response to Comments on the Quality of English Language |
||
5. Additional clarifications |
Round 3
Reviewer 2 Report
I recommend acceptance of the present manucript in its present version
Minor editing of English language required